

# An analysis of abnormalities in the B cell receptor repertoire in patients with systemic sclerosis using high-throughput sequencing

Xiaodong Shi[1,*], Tihong Shao[2,*], Feifei Huo[3], Chenqing Zheng[4], Wanyu Li[5] and Zhenyu Jiang[1]

[1] Rheumatology, First Hospital of Jilin University, Changchun, The People's Republic of China
[2] Rheumatology, The First Affiliated Hospital of Anhui Medical University, Hefei, The People's Republic of China
[3] Intensive Care Unit, First hospital of Jilin university, Changchun, The People's Republic of China
[4] Shenzhen RealOmics (Biotech) Co.Ltd, Shenzhen, The People's Republic of China
[5] Hepatology, First hospital of Jilin university, Changchun, The People's Republic of China
[*] These authors contributed equally to this work.

## ABSTRACT

Systemic sclerosis is a chronic multisystem autoimmune disease that is associated with polyclonal B cell hyperreactivity. The CDR3 of BCRs is the major site of antigen recognition. Therefore, we analyzed the BCR repertoire of patients with SSc. The BCR repertoires in 12 subjects including eight SSc patients and four healthy controls were characterized by high-throughput sequencing, and bioinformatics analysis were studied. The average CDR3 length in the SSc group was significantly shorter. The SSc patient displayed more diverse BCR. Moreover, SSc patients with mild skin sclerosis, anti-Scl70, interstitial lung disease or female sex were more diversified. B cells from the SSc patients showed a differential V and J gene usage. SSc patients had distinct BCR repertoires.These findings reflected the differences of BCR repertoires between SSc patients and controls. The higher-usage genes for the BCR sequence might be potential biomarkers of B cell-targeted therapies or diagnosis for SSc.

Corresponding authors
Wanyu Li, 82232559@qq.com
Zhenyu Jiang, 328430366@qq.com

## INTRODUCTION

Systemic sclerosis (SSc) is a chronic multisystem disease characterized by extensive fibrosis, autoantibody production, and microangiopathy. Fibrosis involves the skin and internal organs, including the lung, gastrointestinal tract, and heart. However, the pathogenesis of SSc is not completely understood (*Sakkas, Chikanza & Platsoucas, 2006*). Recently, B cells have been highlighted as exerting important regulatory effects independent of their antibody-producing function. B cells promote fibrosis by producing autoantibodies, cytokines, and some mediators, and their infiltration in SSc lesions varies (*Hussein et al., 2005*; *Lafyatis et al., 2007*). B cells have also been reported to be involved in the pathogenic

mechanism of the disease. B cells are hyperactive in patients with SSc in the presence of hyper-γ-globulinemia, and increased levels of autoantibodies, free immunoglobulin light chains. (*Lanteri et al., 2014*). In the B cell repertoire of patients with early SSc, 54% of the B cells overexpress the stimulatory receptor CD19, whereas 28% overexpress this receptor in patients with a chronic disease (*Mavropoulos et al., 2016*). The hyperactivity of B cells in SSc patients results in the overexpression of autoantibodies. Some autoantibodies induce a pro-inflammatory and/or fibrotic response in patients with SSc (*Fineschi et al., 2008*; *Walker et al., 2007*). B cell depletion has been induced in patients with SSc through treatment with an anti-CD20 monoclonal antibody (*Jordan et al., 2015*), which not only ameliorated skin fibrosis but also improved/stabilized lung function (*Giuggioli et al., 2015*; *Mahler, Fritzler & Satoh, 2015*; *Villalta et al., 2012*). An in-depth understanding of B cell genetics and biology will provide more strategies and methods for B cell-targeted therapy.

B cells are selectively activated following the specific recognition of antigens by variable regions of surface B cell receptors (BCRs). B cell genes encode the variable regions of antibody heavy and light chain proteins. Further mutation is caused by recombination, junctional diversity, and somatic mutations, resulting in as many as $10^{11}$ unique antibody molecules. Three complementary determinant regions (CDR3s) are present in the variable region of the heavy and light chains that are highly susceptible to somatic hypermutations and encode the amino acid loops of the antigen-binding site. Within CDR3s, the CDR3 region of the heavy chain (VH) plays the main role in determining the specificity and diversity of antibodies and antigens bound (*Giuggioli et al., 2015*; *Glanville et al., 2009*).

The diversity of BCRs in healthy human peripheral blood is estimated to be approximately $3 \times 10^9$, making an analysis of the entire repertoire more difficult(*Xu & Davis, 2000*). Next-generation sequencing (NGS) technology not only enables large-scale DNA sequencing but also enables an in-depth analysis of the BCR repertoire in the circulatory system (*Jung et al., 2006*), thereby facilitating an in-depth study of the diversity and selection mechanism of the BCR repertoire (*Georgiou et al., 2014*).

In the present study, we investigate the BCR repertoire in patients with SSc. The CDRs of BCRs were detected using high-throughput sequencing (HTS) to explore the differences in the BCR repertoire between the SSc and healthy control groups. The large-scale sequencing of this BCR repertoire may improve our understanding of the immune system in patients with SSc. In addition, a deeper understanding of the BCR repertoire in patients SSc will help us understand the mechanisms of B cell hyperactivity and tolerance imbalance in patients with SSc, which we hope will provide a theoretical basis for the development of more effective targeted therapies in the future.

## MATERIALS AND METHODS

### Subjects

In this study, eight patients with SSc conforming to the criteria of the American College of Rheumatology were included (*Calis & Rosenberg, 2014*), and patients with a history of or current use of immunosuppressive therapy were excluded. The eight patients with SSc (SSc group) and four healthy controls (Control group) were treated at the First Hospital

**Table 1  Clinical characteristics of the study sample.**

| ID | Age (years) | Sex | Autoantibodies | Disease duration (years) | Lung involvement | Skin sclerosis degree |
|----|-----|-----|----------------|------------|------|------|
| P1 | 64 | F | ANA | 1 | ILD | Severe |
| P2 | 62 | F | ANA | 1 | ILD | Severe |
| P3 | 65 | F | ANA | 1 | – | Mild |
| P4 | 32 | F | Scl70 | 2 | – | Severe |
| P5 | 58 | F | Scl70 | 1 | ILD | Severe |
| P6 | 63 | F | Scl70 | 4 | – | Mild |
| P7 | 30 | M | Scl70 | 1 | ILD | Mild |
| P8 | 63 | M | ANA | 2 | ILD | Mild |
| H1 | 62 | F | – | – | – | – |
| H2 | 64 | F | – | – | – | – |
| H3 | 65 | F | – | – | – | – |
| H4 | 59 | F | – | – | – | – |

Notes.

P, patients; H, healthy controls; ANA, antinuclear antibody; Scl-70, anti-topoisomerase I antibody; ILD, interstitial lung disease.

of Jilin University, Changchun, China, between January and December 2016 and were enrolled after obtaining informed consent and institutional approval. The patients were diagnosed with diffuse cutaneous SSc based on the criteria reported by *LeRoy et al. (1988)*. Four age-matched healthy subjects comprised the control group. We declare that this study has been approved by the Ethics Committee of the First Hospital of Jilin University (2015–119), and written informed consent was obtained from all participants.

The clinical manifestations and laboratory results were derived from clinical questionnaires. Lung fibrosis was observed using high-resolution computed tomography (HRCT). The modified Rodnan total skin thickness score was used to assess the degree of skin fibrosis (*LeRoy et al., 1988*). The eight patients scored 12–30 points before treatment and were divided into two groups (mild and severe) according to a cutoff for the modified Rodnan score of 18 points. Table 1 shows the clinical and laboratory characteristics of each patient.

## Samples collection

Ten milliliter blood samples were collected, and peripheral blood mononuclear cells (PBMCs) from patients and controls were obtained immediately by density-gradient centrifugation over Ficoll (MD Pacific Biotechnology Co., Ltd, Tianjin, China). Total RNA was extracted from PBMCs by TRIZOL reagent (Invitrogen, Carlsbad, CA,USA)according to the manufacturer's instructions. Nanodrop 2000 was used to evaluate the quantity and purity of RNA, which was reversely transcribed into cDNA.

## Multiplex PCR amplification

The human immunoglobulin heavy chain (IGH) sequences were downloaded from the international ImMunoGeneTics information system (IMGT) (http://www.imgt.org/). A

relatively conserved region 3 was selected as the predicted forward primer region in the upstream sequence of CDR3. A set of primers corresponding to most V gene families were selected. Similarly, 6 reverse primers consistent with J gene families were designed. MFEprimer2.0 (Shanghai, China) and Oligo 7.0 (Colorado Springs, CO, USA) were used to analyze the primers and reverse primers for dimers and loop structures. We cooperated with the BGI-Shenzhen Company to regulate primer concentrations and minimize PCR amplification bias using a new bioinformatics method (IMonitor) (*Clements et al., 1995*; *Liu et al., 2016*). The sequences of low-quality primers were slightly altered. The final primer sequences are shown in Table S1.

The complete VDJ rearrangements of the IGH sequences were amplified from 4ug of DNA using the primers listed in the IMGT database. Each sample was amplified by performing multiple PCRs (Qiagen multiplex PCR kit) with the same amount of DNA using a high-fidelity enzyme. The PCR conditions were 15 min at 95 °C, 25 cycles at 94 °C for 15 s and 60 °C for 3 min, and an incubation at 72 °C for 10 min. PCR products were purified and primer sequences were removed using AMPure XP beads (Brea, CA, USA). Then, the second round of PCR began, and we added a sequencing index to each sample. The PCR conditions were 98 °C for 1 min, 25 cycles at 98 °C for 20 s, 65 °C for 30 sand 72 °C for 30 s, and a final incubation at 72 °C for 5 min. Finally, an agarose gel was used to isolate the target library, and QIAquick gel Extraction Kits (Qiagen) were used to isolate and purify the target region.

## High-throughput sequencing

The Illumina HiSeq sequence adapter was connected and libraries were sequenced using the Illumina HiSeq 4000 platform. The library was quantified using an Agilent 2100 bioanalyzer (Agilent DNA 1000 reagent, Santa Clara, California, USA) and real-time quantitative PCR (TaqMan probes, Shenzhen, China).

## Data analysis

The quality of the Illumina HiSeq 4000 library was evaluated using a BGI formula. Namely, we filtered adapter reads and low-quality reads in the raw data, and further aligned clean data. The clean data were then compared with the human IGH database and analyzed with MiXCR, which categorized the identical and homologous reads into clonotypes and corrected the PCR and sequencing errors using heuristic multi-layer clustering. The obtained data included V, D, J assignments, clustering, CDR3 length distributions and other results.

## Statistical analyses

Statistical analyses were conducted in R software. The Wilcoxon rank-sum test was used to compare the expression level and expression diversity between the SSC and the control group. A two-sided $P < 0.05$ was considered statistically significant.

**Table 2   IGH sequence statistics.**

| Sample ID | Clean fragments | Aligned fragments | Alignment rate (%) | Fragments used | Clonotype number | Out of frame | Reads | Resample reads |
|---|---|---|---|---|---|---|---|---|
| H1 | 7,199,220 | 6,979,148 | 96.94 | 6,157,658 | 26,609 | 9,532 | 6,060,539 | 5,591,443 |
| H2 | 7,619,011 | 7,267,210 | 95.38 | 6,456,957 | 17,265 | 4,620 | 6,367,390 | 5,591,443 |
| H3 | 9,445,266 | 9,130,907 | 96.67 | 8,303,268 | 27,477 | 9,050 | 8,174,162 | 5,591,443 |
| H4 | 8,960,188 | 8,621,604 | 96.22 | 7,612,364 | 25,833 | 8,323 | 7,500,096 | 5,591,443 |
| P1 | 6,218,164 | 6,138,462 | 98.72 | 5,689,474 | 28,377 | 9,157 | 5,591,443 | 5,591,443 |
| P2 | 7,153,755 | 6,986,351 | 97.66 | 6,359,872 | 27,065 | 9,826 | 6,249,078 | 5,591,443 |
| P3 | 7,128,894 | 7,030,004 | 98.61 | 6,508,507 | 25,453 | 8,103 | 6,399,782 | 5,591,443 |
| P4 | 7,689,752 | 7,553,632 | 98.23 | 6,899,318 | 26,620 | 8,669 | 6,774,501 | 5,591,443 |
| P5 | 9,373,266 | 9,271,638 | 98.92 | 8,514,711 | 55,567 | 23,321 | 8,364,642 | 5,591,443 |
| P6 | 9,336,294 | 9,238,211 | 98.95 | 8,465,770 | 46,407 | 14,962 | 8,308,364 | 5,591,443 |
| P7 | 10,881,177 | 1,080,0389 | 99.26 | 9,940,707 | 63,290 | 26,451 | 9,772,110 | 5,591,443 |
| P8 | 9,359,034 | 9,250,877 | 98.84 | 8,464,035 | 49,427 | 20,366 | 8,335,203 | 5,591,443 |

## RESULTS

### Summary of the sequencing results

Eight patients with SSc (SSc group: P1–8) and four healthy controls (Control group: H1–4) were enrolled in this study. After filtering, including the removal of adapter sequences, contamination, and low-quality reads, we acquired an average of 8,363,668 sequencing reads each sample. Table 2 showed the IGH sequence summary (Table 2).

### Distribution of CDR3 lengths

The important determinant of the diversity of the B cell repertoire is the length of the BCR CDR3 loop. In our study, we evaluated the length distribution of the BCR CDR3 sequence (aa) in the SSc and Control groups. The SSc group had a higher percentage of BCR CDR3 sequences of 14 amino acids (aa) in length than the Control group ($P = 0.029$), but a lower percentage of BCR CDR3 sequences of 29 ($P = 0.039$) or 37 ($P = 0.013$) aa in length (Fig. S1). The average CDR3 length was significantly shorter in the SSc group ($17.85 \pm 0.37$ aa) than in the Control group ($18.47 \pm 0.038$ aa; $P = 0.038$) (Fig. S2).

### Degree of expansion and frequency distribution of B cell clones

By aligning and identifying each sequence, we were able to calculate the expression level of each clone. The extent to which each individual clone expands depends on the frequency of the unique CDR3 sequence in a sample. Here, we defined clones with a frequency greater than 0.5% among the analyzed BCRs as highly expanded clones (HECs), clones with a frequency of 0.05–0.5% as high-frequency clones, clones with a frequency of 0.005–0.05% as medium-frequency clones, clones with a frequency of 0.0005–0.005% as low-frequency clones, and clones with a frequency of <0.0005% as rare clones. In terms of the frequency distribution, most of the repertoires were composed of a small amount of HECs distributed in a left-skewed manner, and most clones were present at a low frequency (Fig. S3). In the SSc group, the more highly expanded clones ($9.37 \times 10^{-2} \pm 3.51 \times 10^{-2}\%$) accounted for 10% of the B cell sequences present among the medium-frequency clones (0.005

$-0.05\%$) ($P = 0.005$), and the expanded clones ($1.31 \times 10^{-2} \pm 1.42 \times 10^{-2}\%$) accounted for 2.1% of the B cell sequences present among the high-frequency clones ($0.05 - 0.5\%$) ($P = 0.013$). However, lower expression levels were observed among the rare clones ($0.874 \pm 3.32 \times 10^{-2}\%$) ($P = 0.019$) in the SSc group than in the Control group (Table S2).

## Comparison of the BCR repertoire diversity between groups

We used the Shannon entropy index, which summarizes the frequency of every clonotype, to quantify the BCR repertoire diversity of the SSc group, Control group and subgroups with different features. An analysis of the repertoires of patients with SSc revealed a very high BCR diversity that was higher than the Control group ($P = 0.004$). Moreover, the entire BCR repertoire of the SSc group had a much more diversified clonotype composition than the Control group, particularly in the subgroups with a mild degree of skin sclerosis, anti-Scl70 antibodies, or interstitial lung disease (ILD). Furthermore, the Shannon entropy of the Control group and female and male patients in the SSc group was analyzed; the BCR diversity of female patients was higher than the Control group. Although the $P$ value was not statistically significant, a greater diversity was observed in male patients than in female patients. The sample size of the male group should be increased in subsequent studies (Fig. 1).

## Distribution of similar CDR3 sequences in the SSc and Control groups

The similarity map of the samples was obtained by calculating the distance between each pair of samples as the Jaccard distance $= 1 -$ Jaccard index and constructing the neighbor-joining tree (Fig. 2). According to the map, the subjects were divided into three groups: female patients with SSc (P1–6), male patients with SSc (P7–P8) and healthy controls (H1–4). An obvious difference was detected between the patients and healthy controls, which were well-separated; however, among the patients, the similarities between P7 and P8 were also different from the other patients. (The sequences in male patients with SSc were also different from those in female patients with SSc.)

## Comparison of IGHV and IGHJ repertoires between the SSc and Control groups

The usage features of the V and J gene segments were analyzed for the different clone levels, as shown by the histograms and heat maps, respectively (Figs. 3 and 4), to analyze whether disease-specific differences in the IGHV and IGHJ repertoires existed. The expression levels of the respective IGHV and IGHJ repertoires were compared among the groups. For the IGHV gene segments, IGHV3-9 ($P = 0.04$) were highly expressed in the SSc group. For the IGHJ gene segments, IGHJ4 ($P = 0.02$) showed significantly higher usage in the SSc group. Then, the relative usage frequencies of V–J pairs were compared between the SSc and Control groups (Figs. 3 and 4, Table 3). IGHV1-18-J3,GHV1-8-J2,GHV1-8-J4, IGHV3-53-J4,IGHV3-9-J3,IGHV3-9-J4,IGHV3-9-J5, IGHV4-61-J4, SSc group is higher than Control (Fig. 3C, Fig. 4); and in IGHV2-5-J6,IGHV2-70-J5, IGHV3-20-J6, IGHV3-23-J6, IGHV3-7-J5, IGHV4-4-J6, SSc group is lower than Control (Fig. 4, Table 3).

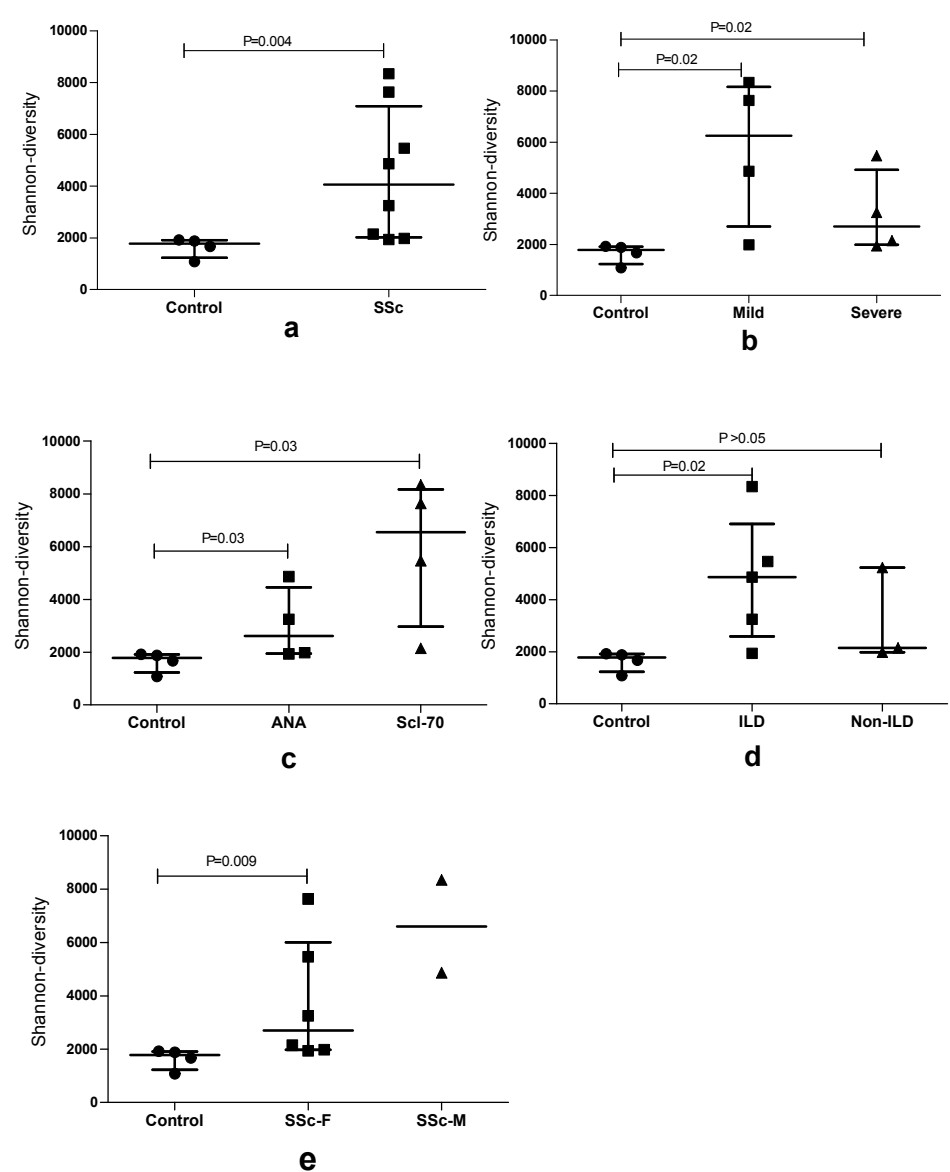

**Figure 1** **The BCR repertoire diversity among SSc patients compared with Control group.** The BCR repertoire diversity among SSc patients compared with the Control group. (A) Diversity between the SSc and Control ($p = 0.004$); (B) Shannon entropy among the mild and severe degrees of skin sclerosis patients compared with the Control group, respectively, ($p = 0.02$); (C) Shannon entropy among antinuclear antibody (ANA)-positive and Scl-70-positive subgroups compared with Control respectively, ($p = 0.03$); (D) Shannon entropy among the ILD ($p = 0.02$) and Non-ILD patients compared with the Control group respectively. (E) Shannon entropy among the female ($p = 0.009$) and male SSc patients compared with Control group. Differences were statistically analyzed for significance by the Wilcoxon rank sum test.

## DISCUSSION

B cells are hyperactivated and produce many autoantibodies in patients with SSc, some of which can lead to collagen production and vasoconstriction (*Zhang et al., 2015*). NGS of the B cell receptor is helping researchers understand and identify important questions in
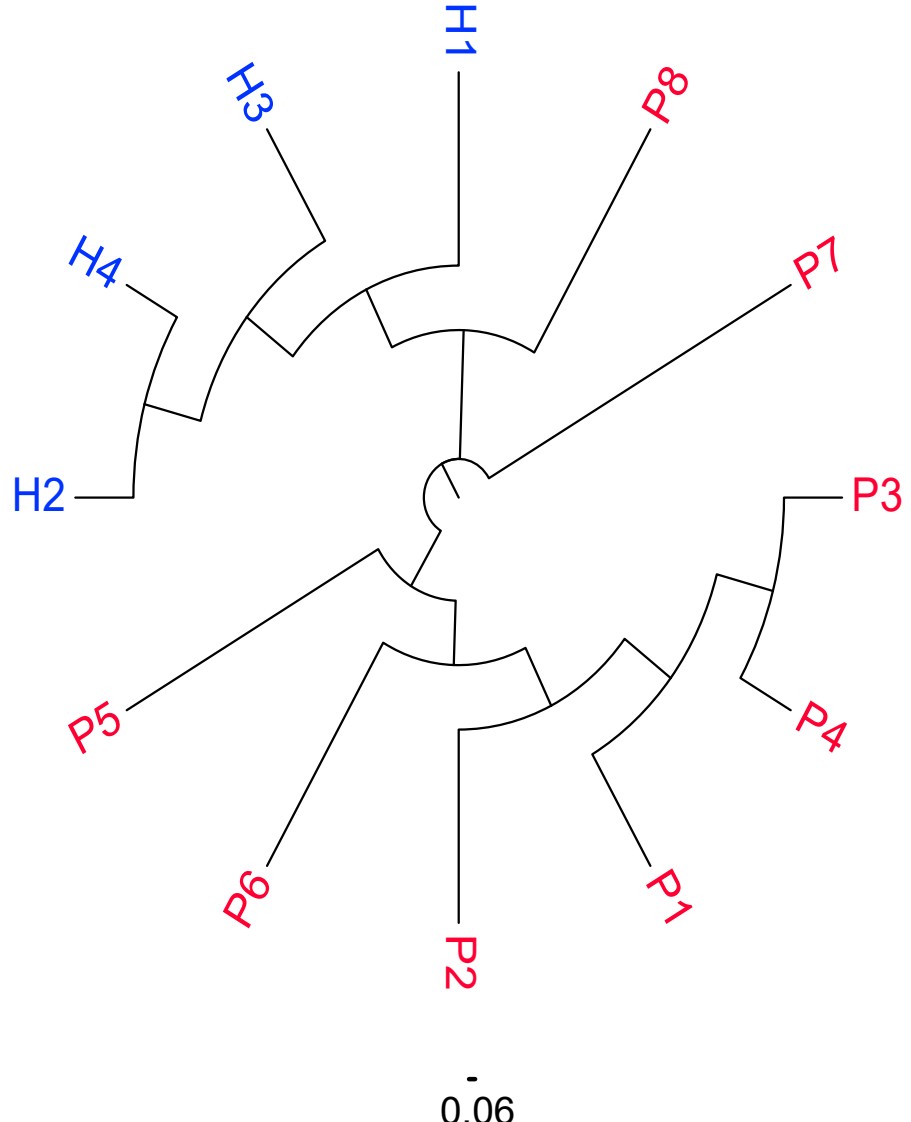

-
0.06

**Figure 2** **Similarity distribution of CDR3 sequences in the SSc and Control groups.** Similarity distribution of CDR3 sequences in the SSc and Control groups by calculating the Jaccard distance between each pair of samples. The subjects were divided into three groups according to the figure: female patients with SSc (P1–6), male patients with SSc (P7–P8) and healthy controls (H1–4).

the field of B cell repertoire development and differences in the B cell subsets in healthy and diseased states. B cells play important roles in various autoimmune diseases, particularly in the occurrence and development of SSc, affecting the immune response of patients to antigens (*Sakkas & Bogdanos, 2016*; *Schanz et al., 2014*). The most variable region of the BCR sequence is the CDR3 region, which directly determines the specificity of antigen binding to the BCR (*Bashford-Rogers, Smith & Thomas, 2018*). HTS allows a detailed investigation of the BCR repertoire. To date, the composition of and variations in the BCR VH CDR3 repertoire of patients with SSc have not been reported. In the present study,

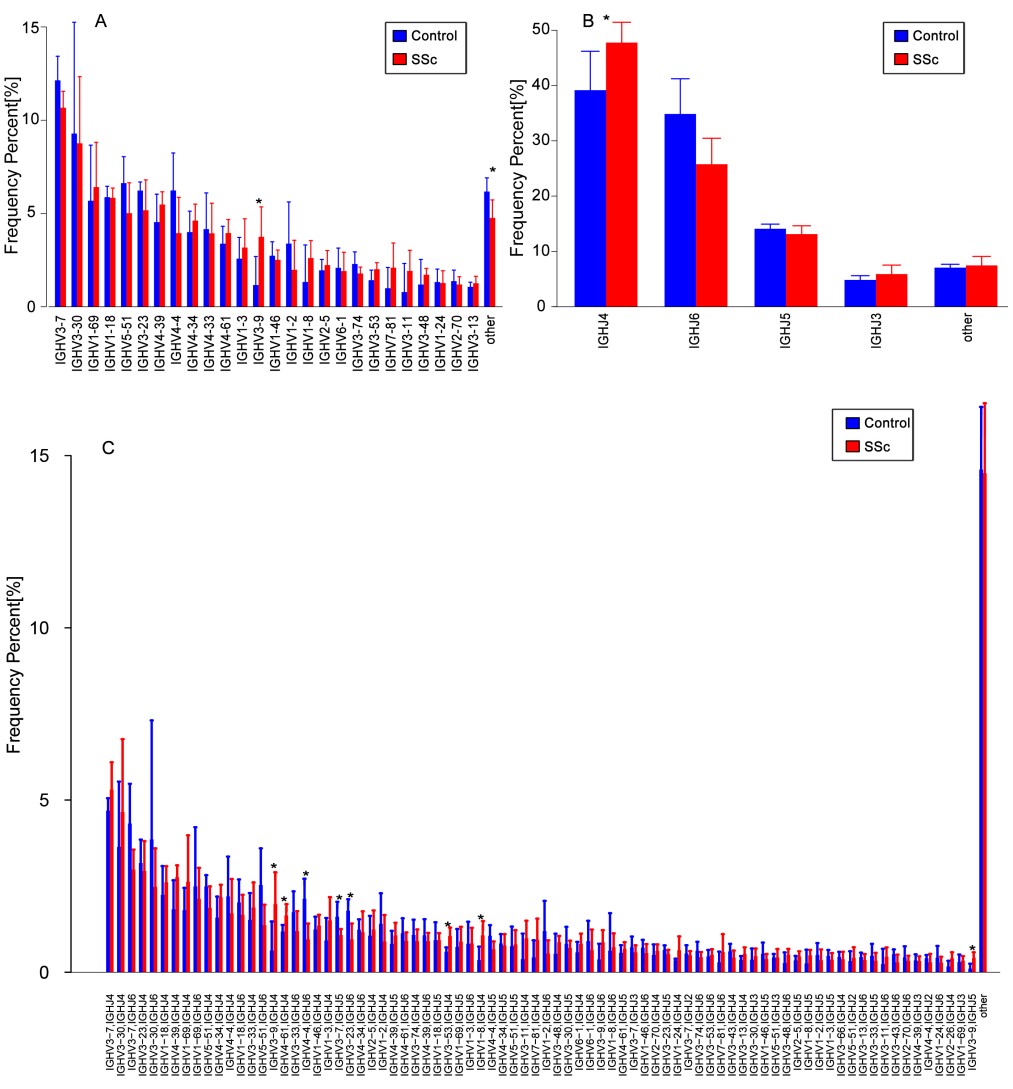

**Figure 3   Relative frequencies of each IGHV gene.** Relative frequencies of each IGHV gene (A) [IGH V3-9: $P = 0.032$ ], IGHJ gene; (B) [IGHJ4: $P = 0.028$] and IGH V-J; (C) [IGHV3-9-J4: $P = 0.041$; IGHV3-9-J5: $P = 0.023$; IGHV4-4-J6: $P = 0.016$; IGHV3-7-J5: $P = 0.045$; IGHV3-23-J6: $P = 0.016$; IGHV3-53-J4: $P = 0.016$; IGHV1-8-J4: $P = 0.016$] fragments in PBMCs from the SSc and Control groups. The bars and error bars indicate the mean frequencies and standard deviations of the results from individual subjects. Differences were statistically analyzed for significance by the Wilcoxon rank sum test (*$P < 0.05$).

we employed a novel NGS method to compare and analyze the repertoire of the BCR IGHV CDR3 region in a group of patients with SSc ($n = 8$) and a group of healthy controls ($n = 4$).

We analyzed the distributions of CDR3 lengths among an average of 8,363,668 filtered sequencing reads per sample, which provided extensive information on the BCR repertoire in the SSc and Control groups. For example, we identified the most frequently observed length. Variable rearrangements cause different CDR3 lengths, the characteristics of BCR clonality are determined by measuring the lengths of CDR3 subsets. However, in our study,
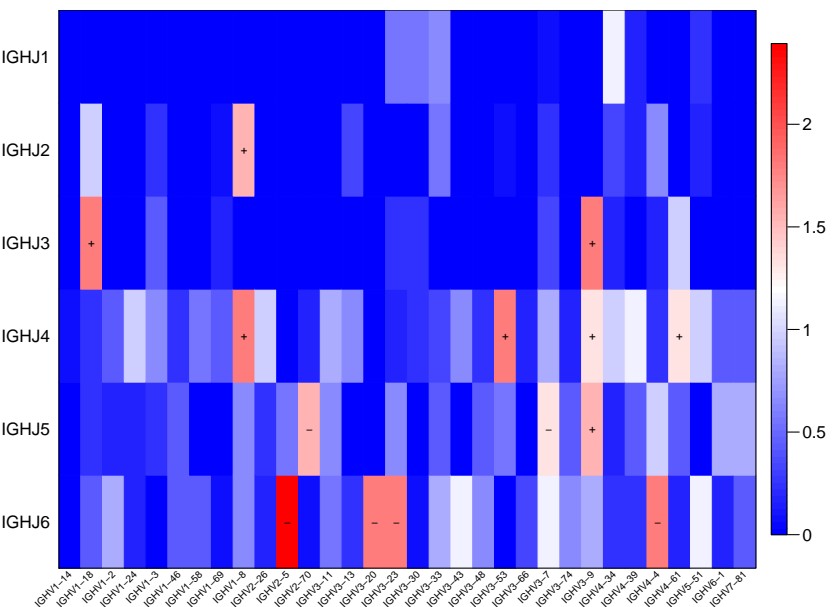

**Figure 4** **Comparison of the average V-J gene utilization of the sequenced IGH sequences between the SSc group and Control group.** Comparison of the average V-J gene utilization of the sequenced IGH sequences between the SSc group and the Control group. The J gene segments are arranged on the $y$-axis, and the V gene segments are arranged on the $x$-axis. The different colors (red to blue rectangular bands) indicate the different levels of significance. Rectangular bands is the value converted from $-\log 10$ ($p$ value), "+" indicates significant difference (SSc group is higher than the Control group), and "−" indicates significant difference (Control group is higher than SSc). Differences were statistically analyzed for significance by the Wilcoxon rank sum test ($p < 0.05$).

**Table 3** **IGHV-J pairing usage differences between SSc and Control groups.**

| IGHV-J pairing | SSc (mean ± s.d)% | Control (mean ± s.d)% | P |
|---|---|---|---|
| IGHV1-18-J3 | 0.35 ± 0.08 | 0.19 ± 0.09 | 0.016 |
| GHV1-8-J2 | 0.16 ± 0.12 | 0.02 ± 0.03 | 0.028 |
| GHV1-8-J4 | 1.07 ± 0.42 | 0.35 ± 0.39 | 0.016 |
| IGHV3-53-J4 | 1.05 ± 0.24 | 0.59 ± 0.13 | 0.016 |
| IGHV3-9-J3 | 0.26 ± 0.09 | 0.04 ± 0.05 | 0.016 |
| IGHV3-9-J4 | 1.98 ± 0.92 | 0.62 ± 0.84 | 0.042 |
| IGHV3-9-J5 | 0.41 ± 0.18 | 0.10 ± 0.14 | 0.028 |
| IGHV4-61-J4 | 1.65 ± 0.33 | 1.18 ± 0.19 | 0.042 |
| IGHV2-5-J6 | 0.22 ± 0.12 | 0.47 ± 0.06 | 0.004 |
| IGHV2-70-J5 | 0.11 ± 0.09 | 0.27 ± 0.11 | 0.028 |
| IGHV3-20-J6 | 0.07 ± 0.06 | 0.36 ± 0.31 | 0.016 |
| IGHV3-23-J6 | 0.95 ± 0.46 | 1.78 ± 0.33 | 0.016 |
| IGHV3-7-J5 | 1.08 ± 0.17 | 1.61 ± 0.43 | 0.042 |
| IGHV4-4-J6 | 0.95 ± 0.46 | 2.13 ± 0.58 | 0.016 |

the distribution of BCR CDR3 lengths showed a higher percentage of BCR CDR3 sequences of 14 aa in length in the SSc group than in the Control group ($P = 0.029$), but a lower percentage of BCR CDR3 sequences of 29 aa ($P = 0.039$) or 37 aa ($P = 0.013$) in length in the SSc group. In addition, the average CDR3 length in the SSc group ($17.85 \pm 0.37$ aa) was significantly shorter than in the Control group ($18.47 \pm 0.038$ aa; $P = 0.038$). Interestingly, the length of the BCR CDR3 is also significantly shorter in patients with systemic lupus erythematosus (SLE) than in the controls (*Robins, 2013*). However, this effect may be due to the continuous increase in the proportion of plasmablasts in patients with SLE, since the CDR3 length of BCRs on naïve B cells is often longer than on B cells with antigen experience (*Liu et al., 2017*). Moreover, B cell clones from patients with rheumatoid arthritis (RA) are enriched for longer heavy chain CDR3 lengths (*Doorenspleet et al., 2014*; *Galson et al., 2015*; *Samuels et al., 2005*). In the present study, we mainly observed the B cell repertoire of patients with SSc and healthy controls, and found that the number of different antibody types and B cell subsets were not well differentiated. Based on our results, patients with SSc have CDR3 regions with shorter lengths in the overall repertoire. Thus, the repertoire of different B cell subsets in patients with SSc requires further research.

The Shannon entropy index and the distribution of HECs differed between the two groups. Most individual B cell clones occurred at very low frequencies, suggesting that these clones have not yet expanded. However, the expansion of clones was observed in both groups; among medium-frequency clones, a greater degree of expansion was observed in the SSc group than in Control group (0.005–0.05%) ($P = 0.005$). One possible explanation for this difference is that the existence of autoantigens induces the development of new autoreactive clones.

The diversity analysis revealed that the repertoires of patients with SSc exhibited a much higher BCR diversity than the Control group ($P = 0.004$). Moreover, the clonotype composition of the BCR repertoire was much more diversified in patients with SSc presenting a mild degree of skin sclerosis, anti-Scl70 antibodies, ILD or the female sex than in the controls. As shown in the study by *Forestier et al. (2018)*, B cells undergo clonal expansion in response to chronic stimulation, which is potentially caused by autoantigens or pathogens. In addition, variations in IGHV genes are associated with disease susceptibility. Both patients with SSc and tight-skin mice (the genetic model of SSc) display intrinsic B cell abnormalities that are primarily characterized by chronic B cell activation (*Brezinschek et al., 1997*; *Hasegawa, 2010*; *Saito et al., 2002*). Based on our results, the higher BCR diversity might be associated with the occurrence and development of SSc.

Additionally, although the size of males of SSc patients was small, and it was still not statistically comparable with the female and control groups, the diversity of male SSc patients tends to increase according to the data (Fig. 1E). Consistent with this finding, a noticeable difference in the distribution of similar CDR3 sequences was observed between patients and healthy controls by calculating the distance between each pair of samples (Fig. 2). Furthermore, among the patients, the similarity of the male patients (P7 and P8) was obviously different from the female patients. Sex hormones may play a role in sex-differences in autoimmune diseases. Most human autoimmune diseases, like systemic lupus erythematosus (SLE), Sjogren's syndrome (SS) etc., have increased incidence and

prevalence in females (*Lee & Chiang, 2012*), a study has shown that examination of the B cell subsets in estrogen-treated groups revealed the reduction in the number of transitional B cells, B cell lymphopoiesis is reduced in both pregnant and in the estrogen-treated population (*Bynoe, Grimaldi & Diamond, 2000*). In our study, the diversity of the BCR repertoire in male patients with SSc is different from female patients, It may be that the B cells subsets, growth and activation states and immune response in sex hormones differ. We need to increase the sample size of the male patient to further confirm a different state of the male human antibody repertoire in systemic sclerosis.

BCR repertoires in patients with SSc showed significant changes in IGHV gene usage compared to healthy controls. Our results showed an enrichment of IGHV3-9 and IGHJ4 gene family usage, specificallyIGHV1-18-J3, GHV1-8-J2, GHV1-8-J4, IGHV3-53-J4, IGHV3-9-J3, IGHV3-9-J4, IGHV3-9-J5 and IGHV4-61-J4 usage in patients with SSc was higher than Control, but in IGHV2-5-J6, IGHV2-70-J5, IGHV3-20-J6, IGHV3-23-J6, IGHV3-7-J5, IGHV4-4-J6, the usage was lower than Control. The usage of these pairs was significantly different between the SSc and Control groups (Figs. 3 and 4 and Table 3). These enrichments, particularly the usage of IGHV1 and IGHV4 and the IGHV3, IGHJ4, and IGHJ6 combinations, have also been observed in studies of multiple autoimmune diseases (*Arbuckle et al., 2003*; *Odendahl et al., 2000*; *Shi et al., 2016*; *Tipton et al., 2015*). Furthermore, our study has shown that the usage of IGHV2-5-J6 in SSc was significantly lower than control, which is consistent with de Bourcy's study, IGHV2-5 was used at a lower level in SSc-PAH participants (*De Bourcy et al., 2017*). It has been proved that B cells changes of systemic sclerosis is related to the balance in naive and memory B cell, and that has been shown, an imbalance in naive/switched and unswitched memory B cells that may explain the B cell repertoire abnormality observed in our study (*Simon et al., 2016*). In the study, the higher-usage genes also provide additional information for the future study of effective B cell-targeted therapy or prognostic and/or diagnostic biomarkers for SSc.

In conclusion, the analysis of the BCR immune repertoires of patients with SSc using this meaningful method has an important application value for studying the prognosis and evaluating the clinical responses to treatment in the future. Although this study has some limitations due to the small sample size, future investigations aimed at improving our understanding of the role of the BCR repertoire in immune responses, autoimmunity and autoreactivity are anticipated as the cost of HTS decreases: first to analyze the repertoires of different B cell subsets in patients with SSc, and second to further identify the features of the BCR repertoire and the functions of specific BCR genes in different patients with SSc.

### Funding
This work was supported by the Science and Technology Plan Foundation of Jilin Province No. 20160101008JC. The funders had no role in study design, data collection and analysis, decision to publish, or preparation of the manuscript.

## Grant Disclosures

The following grant information was disclosed by the authors:
Science and Technology Plan Foundation of Jilin Province: 20160101008JC.

## Competing Interests

The authors declare there are no competing interests. Chenqing Zheng is employed by Shenzhen RealOmics (Biotech) Co.Ltd.

## Author Contributions

- Xiaodong Shi conceived and designed the experiments, performed the experiments, prepared figures and/or tables, and approved the final draft.
- Tihong Shao conceived and designed the experiments, performed the experiments, analyzed the data, prepared figures and/or tables, authored or reviewed drafts of the paper, and approved the final draft.
- Feifei Huo analyzed the data, prepared figures and/or tables, and approved the final draft.
- Chenqing Zheng analyzed the data, prepared figures and/or tables, authored or reviewed drafts of the paper, and approved the final draft.
- Wanyu Li conceived and designed the experiments, authored or reviewed drafts of the paper, and approved the final draft.
- Zhenyu Jiang conceived and designed the experiments, authored or reviewed drafts of the paper, and approved the final draft.

## Human Ethics

The following information was supplied relating to ethical approvals (i.e., approving body and any reference numbers):

Ethics Committee of the First Hospital of Jilin University approved the study (2015-119).

## Data Availability

Data is available at Figshare: Shi, Xiaodong (2019): Analysis report.tar.gz. figshare. Dataset. DOI: 10.6084/m9.figshare.9816224.v2.

Shi, Xiaodong (2019): Clean data Results (BCR). figshare. Dataset. DOI: 10.6084/m9.figshare.10279349.v2.

## Supplemental Information

Supplemental information for this article can be found online at http://dx.doi.org/10.7717/peerj.8370#supplemental-information.

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
