# Peer review of "An analysis of abnormalities in the B cell receptor repertoire in patients with systemic sclerosis using high-throughput sequencing"

_PeerJ, doi:10.7717/peerj.8370_

## Round 0.1 · original submission · Major Revisions

Please respond to all the reviewers's comments.

It is essential that the manuscript is edited for typographical errors. Please note that the reviewers have mentioned only some of them. For instance, "gDNA" in line 94 seems a typographical error.

The clarity needs to be improved too. For instance, lines 135-139 are challenging to understand. In figure 4, the p-value thresholds are unclear.
The reporting of the statistical analysis is incomplete.

It is not clear how the statistical analysis of the data was performed in particular concerning multiple comparisons. The fact that male and female patients display different trends is a concern, as the control group only includes females. Please consider performing all your the analysis on female individuals only.

Both reviewers have expressed doubts about the statistics. In the materials and methods section, please explain in which type of comparison each statistical test (Bonferroni, student's, wilcoxon's) was used, briefly explaining the rationale for the choice of each test, and providing references of other BCR repertoire studies analyzed with the same statistical approach.

Please, in the "sample collection" section, specify the volume of blood used (for each individual, if different among individuals).

It would be interesting to add an analysis of the public (shared) clonotypes.

Reviewer 1 ·

Basic reporting

In this manuscript, Shi et al analyze the B cell receptor repertoire in SSc patients using high-throughput sequencing. The manuscript is well-written and easy to read, here are some comments to strengthen the manuscript.

- refer to Control as 'Control' and not Con, as that is not obvious to the reader.
- Figure 1 legend needs to be edited- currently includes random characters.
- Figure 2 legend needs to elaborate to make it clear to the reader.
- Introduction, line 40- 'SSc patients'
- In figure 1, statistical test comparing SSc-Males to controls and Non-ILD to ILD are missing. If the tests are not significant, it must be indicated in the figure.
- Is the Wilcoxon test one-sided or two-sided?
- In each figure legend, please clearly indicate what statistical test was used.

Experimental design

- The authors need to frame their research questions in context of previously published work by de Bourcy et al, 2019, Science Immunology. Dynamics of the human antibody repertoire after B cell depletion in systemic sclerosis. This important work relevant to this manuscript has not been discussed.
- One additional method of stratification to include is based on expression of other SSc-specific autoantibodies, if the authors have access to the data.

Validity of the findings

- Can the authors compare their findings to previously published work by de Bourcy et al?
- Why is it that patients with mild disease display more BCR diversity compared to with severe disease? Also, why is the BCR more diverse in SSc-Males compared to SSc-Females- is this difference statistically significant?

Reviewer 2 ·

Basic reporting

I have only few comments
1- Concerning the form manuscript should be improved by correcting typographical error.
For example :
- line 41 the reference is incomplete: the year is missing
-line 82 Ficoll and not Ficol
- line 129 P=0.039 and not P 0.039
2- All the figures quality must be improved:
- Figure 1: number of patients and the statistic test are not indicated
- figure 2: the legend must be develop to explain how to understand the result
- Figure 3 C: almost unreadible

Experimental design

The analysis of the similarity should be more detailled in the text as well as in the figure legend

Validity of the findings

There is few results concerning B cells repertoire in systemic sclerosis and the development of new technologies using high-throughput sequencing gives the opportunity to investigate the question.
1- My major comment concerning the paper is relative to the small number of patient included in the study that may not be enough to support the conclusions. There are only 8 patients and 4 controls which are not totally paired on sex and age. Moreover, the 8 patients are divided in subgroup of 4. I have strong doubts about the robustness of the results. Differences are observed between male and female in patient groups but there is only female in the controls!
2- All the statistics are done using Wilcoxon test, that should be justified by the authors since patients are not paired.
3- It has been shown in systemic sclerosis, an inbalance in naive/switched and unswitched memory B cells that can explained the B cell repertoire abnormality observed by the authors. I understand that it could be difficult to sort these population in the patients nevertheless the authors should discuss this point (Simon et al, Clin Exp Rheumatol. 2016)

Additional comments

In the manuscipt untitled “An analysis of abnormalities in the B cell receptor repertoire
in patients with systemic sclerosis using high-throughput sequencing” Xiaodong Shi & al analysed the B cell repertoire by high throughput technology. The analysis of the BCR immune repertoires of patients with SSc using new technology has important application value for classification of the patients and treatment efficacy evaluation. Nevertheless, as it was already specify the weakness of this work is the low number of patients and controls

---

## Round 0.2 · Minor Revisions

Please revise according to the reviewer's advice and correct all spelling and grammar and reference mistakes.
I also noticed some mistakes and I suggest the following changes:

46-47: The SSc patients displayed a more diverse BCR
49: these findings reflected the differences in BCR repertoires
74: CDR3 region of the heavy chain (VH) plays the main role
118: (Qiagen multiplex PCR kit)
118: with the same
128: real-time
136: the Wilcoxon rank-sum test
136: was used to compare
196: B cell subsets in healthy and diseased states
198: The most variable region of the BCR
220: However, the expansion of clones was
238: etc.,
242: We need to increase the sample size of the male patient to further confirm a different
252: Furthermore, our study has shown that the usage
258: has an important application value

Reviewer 2 ·

Basic reporting

The manuscript by Shy et al has been improved as previously required but there is still few tipographical or spelling errors to correct
ex: in the discussion "imbalance" and not "inbalance"
There is an error concerning a reference the discussion that need to be corrected:
"As shown in the study by Brezinschek et al. (Forestier et al. 2018)",

Experimental design

No comment

Validity of the findings

Some points has been clarify compared to the first version of the paper (especially the statistics).
Concerning the small number of patients and the difference observed between male and female patients the autours have relativized their result and underlined that this is a pilot study .

---

## Round 0.3 · accepted · Accept

Thank you for your submission.

Kind regards,
Antonella Prisco